

# The climatology of planetary boundary layer height in China derived from radiosonde and reanalysis data

Jianping Guo[1*], Yucong Miao[1], Yong Zhang[2], Huan Liu[1], Zhanqing Li[3, 4 *], Wanchun Zhang[1], Jing He[1], Mengyun Lou[1], Yan Yan[1], Lingen Bian[1], Panmao Zhai[1]

[1]State Key Laboratory of Severe Weather & Key Laboratory of Atmospheric Chemistry of CMA, Chinese Academy of Meteorological Sciences, Beijing 100081, China
[2]Meteorological Observation Centre, China Meteorological Administration, Beijing 100081, China
[3]College of Global Change and Earth System Science, Beijing Normal University, Beijing 100875, China
[4]Dept. of Atmospheric & Oceanic Sciences and ESSIC, University of Maryland, College Park, MD 20740, USA

*Correspondence to*: Drs. Jianping Guo (jpguocams@gmail.com) and Zhanqing Li (zli@atmos.umd.edu)

**Abstract.** The important roles of planetary boundary layer (PBL) in climate, weather and air quality have long been recognized, but little has been known about the PBL climatology in China. Using the
fine-resolution sounding observations made across China and a reanalysis data, we conducted a comprehensive investigation of the PBL in China from January 2011 to July 2015. The boundary layer height (BLH) is found to be generally higher in spring and summer than that in fall and winter. The comparison of seasonally averaged BLH derived from observations and reanalysis shows good agreement. The BLH derived from three- or four-times-daily soundings in summer tends to peak in the
early afternoon, and the diurnal amplitude of BLH is higher in the northern and western sub-regions of China than other sub-regions. The meteorological influence on the annual cycle of BLH are investigated as well, showing that BLH at most sounding sites is negatively associated with the surface pressure and





lower tropospheric stability, but positively associated with the near-surface wind speed and temperature. This indicates that meteorology plays a significant role in the PBL processes. Overall, the key findings obtained from this study lay a solid foundation for us to gain a deep insight into the fundamentals of PBL in China, which helps understand the roles of PBL playing in the air pollution, weather and climate

of China.

## 1. Introduction

The planetary boundary layer (PBL), the lowest layer of troposphere, is directly influenced by the

Earth's surface, at a response timescale of about an hour or less (Garratt, 1994; Stull, 1988). The accurate parameterization scheme with regard to how the PBL evolves is critical to prediction of weather, climate and air quality, which heavily relies on the high resolution observations of atmospheric profiles (Hu et al., 2010; Wood, 2012; Xie et al., 2012). Through the PBL, the exchanges of momentum, water, heat and air pollutants take place between the Earth's surface and the free troposphere. To

characterize the structure of PBL, the concept of boundary layer height (BLH) is commonly used (Seibert, 2000; Seidel et al., 2010), since the BLH determines the vertical extent of turbulent mixing, vertical diffusion, and convective transport within PBL.

The development of PBL is dominated by the complex surface forcings, including sensible heat flux, frictional drag, evaporation, transpiration, and terrain induced flow modification (Stull, 1988). As a

result, the BLH is quite variable both in time and space. During a diurnal cycle, the BLH is typically





shallow (a few hundred meters) at night due to the strong near-surface stability, and the PBL can be well developed and reach several kilometers in the afternoon.

Traditionally, the BLH is diagnosed using the height-resolved observations of temperature, humidity, and wind from radiosondes (Seibert, 2000; Seidel et al., 2010). The vertical resolution of these profiles

are not sufficient to estimate BLH, since most soundings only provide a few records below 500 hPa (Liu and Liang, 2010). Partly due to the lack of fine-resolution soundings, there have been rather limited investigations of the BLH climatology over specific locations and regions using atmospheric sounding data (Liu and Liang, 2010; Norton and Hoidale, 1976; Seidel et al., 2012).

In addition to the traditional radionsonde-based PBLH algorithms, other methods relying on new data

sources such as ground-based lidar (Hennemuth and Lammert, 2006; Sawyer and Li, 2013), sodar (Beyrich, 1997), ceilometer (Eresmaa et al., 2006), aircraft sounding (Dai et al., 2014), space-borne lidar (Chan and Wood, 2013; Zhang et al., 2016), have been put forward to determine the PBL structure and its processes. However, the multiple types of observations tend to reveal different aspects of the PBL characteristics, due to the differing atmospheric variables used, leading to inevitable BLH

differences (Seidel et al., 2012). Besides, even for a single data source, the use of different methods to estimate BLH could yield a wide range of results (Seidel et al., 2010). If the same approach is applied to different types of measurements, such disagreements will be lowered (Sawyer and Li, 2013). Therefore, it is better to employ a consistent method to a single type of observation to construct the BLH climatology.

The BLH climatology has been highly sought in the PBL community, pioneered by the research of Holzworth (1964), who used early radiosonde data to study BLH over the U.S., followed by the work of



the U.S. Environmental Protection Agency Air Quality Modeling Group, who makes the U.S. mixing height data product publicly accessed (available at http://www.epa.gov/scram001/mixingheightdata.htm), but it only limited to the 1980s and thereafter. By extending the radiosonde data, new aspects of the BLH climatology over typical regions have been

further investigated, producing a much clearer regional picture of BLH (Liu and Liang 2010; Seidel et al. 2010; Sawyer and Li, 2013). A recent climatological BLH studies (Seidel et al. 2012) gained new insight into the diurnal and seasonal variations of BLH over the continental United States and Europe by combining the radiosonde observations, a reanalysis and two climate models.

There are a host of literatures (e.g., Zhang et al., 2014; Tang et al., 2016) attempted to elucidate BLH
characteristics in China, but most of them are limited to a specific place, let alone the BLH climatologies across China. They did not give much information of PBL over China, in part due to the lack of high-resolution observations in China priori to 2011. After then, the radiosonde data have been acquired at a much higher vertical resolution (1 second resolution) across China by the China Meteorological Administration (CMA), rendering a unique opportunity to study the PBL features across
China.

Using a consistent method and a traditional data source, radiosonde observations, BLH climatologies are presented in this study, the three and a half years of (from January 2011 to July 2015) sounding data acquired by the CMA sounding network, in conjunction with a modern reanalysis data product, are used to investigate the PBL climatology in China. To our knowledge, this is the first observation-based
nation-wide PBL climatology in China. It provides us a unique opportunity to explore the relationship between PBL structure and other meteorological parameters in China.



The main goals of this study are to investigate the BLH climatology in China, and to use it to explore the potential influence of meteorology on the climatological BLHs. Section 2 describes the data and method used, followed by the uncertainty analysis for the BLH retrievals in section 3. Section 4 presents the climatology of BLH in China, as well as the associations with atmospheric variables.

Finally, the key findings are summarized in section 5.

## 2. Data and methods

The radiosonde network of L-band sounding system dated back to 2002 when the China Meteorological Administration (CMA) began upgrading its radiosonde system. As of the beginning of

2011, deployment of the L-band sounding systems have been expanded to 121operational radiosonde stations (Figure 1). The GTS1 digital electronic radiosonde, one of key components of the L-band sounding system, is now widely used in operational radiosonde stations in China, providing fine-resolution profiles of temperature, pressure, relative humidity, wind speed and direction twice a day at 0000 (0800) and 1200 UTC (2000 BJT) (Figure 2).

Previous intercomparison studies (e.g., Bian et al., 2011) indicate that the two types of radiosondes, i.e., Vaisala RS80 and GTS1, agree very well in the profile measurements in the troposphere (including PBL), albeit the large biases in the upper atmospheric levels. Therefore, the data from radiosondes of GTS1 are good enough to derive BLH for the superior performance in PBL.

The sounding data collected from January 1, 2011 to July 31, 2015 are used here to obtain the BLH at

widely scattered radiosonde network across China. In total, we have made use of 391,552 profiles



across China, including 1,578 profiles at 0200 BJT, 190,027 profiles at 0800 BJT, 10,313 profiles at 1400 BJT, and 189,634 profiles at 2000 BJT. In summer (the monsoon season), up to two additional soundings are launched occasionally at 0600 UTC (1400 BJT) and 1800 UTC (0200 BJT) during certain intensive observing periods at selected stations. The fine-resolution sounding observations provide

essential information to investigate the diurnal structure of PBL.

In addition to the CMA sounding observations, the simultaneous profiling data of temperature and wind from ERA-Interim reanalysis (Dee et al., 2011) are also used to derive BLH in this study. The ERA-Interim reanalysis assimilates a variety of measurements, including radiosonde observations at highly discrete levels (the so-called standard sounding levels), into weather prediction models in a

physically consistent manner. The horizontal resolution of ERA-Interim reanalysis is 0.7 degree. In the vertical, there are 60 layers using hybrid coordinates that extend from the surface to the top of 0.1 hPa level, with 21 layers between the surface and 5 km above ground level (AGL). The vertical resolution is about 20 m near the surface, and gradually decreases to about 200 m at 900 hPa level and 500 m at 500 hPa level.

Similar to the methods used by Seidel et al. (2012) to investigate the characteristics of BLH climatology in the United States and Europe, the bulk Richardson number (Ri) method (Vogelezang and Holtslag, 1996) was taken to simultaneously estimate the BLH from CMA soundings and ERA-Interim data. The Ri method has been proved to be one of the best methods for BLH climatology analysis, since it is suitable for both stable and convective boundary layers and can be applied to large amount of

radiosonde and reanalysis data (Seidel et al., 2012). Ri is defined as the ratio of turbulence associated with buoyancy to that induced by mechanical shear, which is expressed as





$$\mathrm{Ri}(z) = \frac{(g/\theta_{vs})(\theta_{vz}-\theta_{vs})(z-z_s)}{(u_z-u_s)^2+(v_z-v_s)^2+(bu_*^2)} \qquad (1)$$

where z denotes height above ground, s the surface, g the acceleration of gravity, $\theta_v$ virtual potential temperature, u and v are the component of wind speed, and $u_*$ the surface friction velocity. $u_*$ can be ignored here due to the much smaller magnitude compared with bulk wind shear term in the denominator (Vogelezang and Holtslag, 1996). Similar to the study of Seidel et al. (2012), the lowest level z at which interpolated Ri crosses the critical value of 0.25, is referred to as BLH. A case in point for the BLH derived from sounding profiles in Beijing is presented in Figure 2. For purposes of simplicity and clarity, the BLHs derived from CMA soundings and ERA-Interim reanalysis are referred to as CMA-BLH and ERA-BLH hereafter, respectively. Unless noted otherwise, all the values of BLH are presented with reference to the height above ground level, rather than that above sea level, to rule out the impact of topographic variation.

For the intercomparision between ERA-BLH and CMA-BLH, all profiles from reanalysis have been sampled at 9 grid points centered at the radiosonde locations. Meanwhile, the ERA-Interim reanalysis data were sampled at 0000 (0800), 0600 (1400), 1200 (2000), and 1800 UTC (0200 BJT) to match the radiosonde observation times. Using instantaneous BLH estimates from the radiosondes and ERA-Interim, we computed seasonal-averaged (DJF, MAM, JJA, and SON) 5th, 25th, 50th, 75th, and 95th percentile BLH values for each station or grid point investigated, separately for 0200, 0800, 1400, and 2000 BJT.

## 3. Uncertainty analysis



Since the selection of critical Ri may bring uncertainty to the estimated BLH from the bulk Ri method as shown in Eq. (1), one case study has been carried out based on the 2000 BJT 03 August 2012 radiosonde observation at Beijing site. The BLHs for differing critical values of 0.2, 0.25, and 0.3 were then derived in attempt to quantify the uncertainties. As illustrated in Figures 2, the difference among the BLHs computed using different critical Ri can be hardly recognized, although the BLH of Ri = 0.3 seems slightly higher.

In the spirit of overall evaluation of the ensemble BLH estimations, scatter plots have made to show CMA-BLHs computed using 0.2 and 0.25 as critical Ri, versus using 0.25 and 0.3 as critical Ri. As expected, the CMA-BLHs of Ri = 0.3 (0.2) are generally higher (lower) than those of Ri = 0.25, however, the CMA-BLH based on various Ri is significantly correlated with each other (R = 0.99). Figures 3c and 3d present the uncertainties as a function of BLHs (Ri = 0.25). It is found that the 50th and 75th percentile values of the absolute uncertainties are < 0.05 km and < 0.1 km, respectively; and the 50th and 75th percentile values of the associated relative uncertainties are both < 5% for BLH (Ri = 0.25) > 2 km, and < 15 % for BLH (Ri = 0.25) < 2 km. As such, the uncertainty caused by the selection of critical value is quite small.

## 4. Results and discussion

This section presents the basic BLH climatologies from the CMA radiosondes and ERA-Interim, including seasonal and diurnal variations. Intercomparisons of climatological results from radiosonde observations with the ERA-Interim are made to better elucidate the BLH characteristics in China. Finally we will explore the potential influences of meteorology on the observed annual cycle of BLH.





*4.1 Overall climatological pattern*

Figures 4a-b present the frequency distributions of CMA-BLH at 0800 and 2000 BJT. At 0800 BJT, almost all the CMA-BLHs are lower than 1 km, with a mean value of merely 0.19 km. The CMA-BLHs at 0800 BJT are typically lower than those at 2000 BJT.

The frequency distributions of summertime CMA-BLH at four observed times (0200, 0800, 1400, and 2000 BJT) are also illustrated in Figures 4c-d. During a diurnal cycle, the CMA-BLH peaks in the afternoon (1400 BJT), with a mean value of 1.25 km. After sunset, the daytime convective boundary layer undergoes a transition into the nocturnal stable boundary layer. The soundings of 2000 BJT provide information about the transition state, and the soundings of 0200 BJT capture the nocturnal boundary layer. In the next morning (around 0800 BJT), after sunrise the PBL experiences another transition to the convective boundary layer.

*4.2 Seasonal variation of CMA-BLH and ERA-BLH and their intercomparison*

Since only the soundings of 0800 and 2000 BJT are launched conventionally, the seasonal variation of BLH is mainly investigated at these two moments (Figure 5). At 2000 BJT, the CMA-BLH generally follows a downward trend from spring to winter (Figure 5b), that is, spring > summer > fall > winter. The climatologically strongest near-surface wind speed in spring (Guo et al., 2011; Zhao et al., 2009) and the intense solar radiation in summer (Miao et al., 2012, 2015), which favor the development of boundary layer during these two seasons. In spring, the mean value of CMA-BLH is ~0.71 km, which is



slightly higher than that of summer (~0.65 km). And the CMA-BLH could occasionally excess 2 km (95th percentile value) in spring and summer.

In contrast, in fall and winter, the mean values of CMA-BLH are merely ~0.32 km, which are significantly lower than those of spring and summer. Such a huge difference of the seasonally averaged
CMA-BLH between the warm seasons (spring and summer) and cold seasons (fall and winter) at 2000 BJT is well captured by the ERA-BLHs, although the ERA-BLHs tend to underestimate the springtime BLHs, and overestimate that of summer (Figure 5b). The scatter plots of seasonally averaged CMA-BLHs and ERA-BLHs (Figure 6), demonstrate the close correspondence between these two kinds of BLH, and give confidence that the seasonally averaged ERA-BLHs can be used as an alternative when
the CMA soundings are unavailable.

In terms of the spatial distributions of BLHs, there exists solar radiance difference received at various stations or grid points across China due to the large spatial range. For a sounding observation at a given time (e.g., 0200, 0800, 1400, 2000 BJT), zonal evolution of BLHs have to be considered from the perspective of actual solar radiance received at the given area.

The spatial distributions of seasonally averaged CMA-BLHs and ERA-BLHs at 0800 BJT are presented in Figure 7. At 0800 BJT, due to the solar radiation in eastern China is stronger than that of western regions, higher BLHs can be distinctly seen over the eastern regions (Figure 7). This BLH spatial pattern is more prominent in spring and summer than in fall and winter (Figures 7a-b). Comparing with the BLHs at 2000 BJT, it is found that the BLHs of 0800 BJT are generally lower
(Figures 4a and 5), and the seasonal variation of BLH is weaker. Despite these differences, the BLHs of 0800 BJT show a similar seasonal variation, that is, higher BLHs are in spring and summer.



As illustrated in Figure 8, both CMA-BLHs and ERA-BLHs at 2000 BJT demonstrate a strong east-to-west BLH gradient in the warm seasons (spring and summer), with higher BLHs over the western China (Figures 8a-b). The BLH spatial pattern may be caused by the different solar radiation in the west (stronger radiation at an earlier Local Sidereal Time) and east of China at 2000 BJT.

In fall, it is interestingly to note that the east-to-west gradient of BLH is less prominent (Figure 8c). Likewise, the BLHs during winter in most regions of China are less than 0.3 km, except for the Tibet Plateau, in which the BLHs can exceed ~0.5 km at 2000 BJT.

Since the soundings of 0200 and 1400 BJT are only launched in summer, the seasonal variation of BLH at these two moments cannot be investigated by using the CMA-BLH alone. Therefore, the
seasonal variations of BLH at 0200 and 1400 BJT are roughly evaluated by using the seasonally averaged ERA-BLHs. It is found that at 1400 BJT the ERA-BLHs in spring and summer are higher than those of fall and winter (Figure 9). At 0200 BJT, most plain regions of China are covered by a relatively shallow boundary layer, without significant seasonal variation (Figure 10).

*4.3 Diurnal variation of BLH*

In summer, four times per day (i.e. 0200, 0800, 1400, and 2000 BJT) soundings are available at most of the radiosonde sites, which allows us to investigate the summertime diurnal variation of BLH (Figure 11a).

With the intense solar radiation in summer, the boundary layer is more fully developed during the
daytime. At 1400 BJT, more than half of the seasonally averaged CMA-BLHs are higher than 1 km (c.f.,





Figures 9b and 11a). Both the CMA-BLH and ERA-BLH demonstrate a south-to-north gradient of BLH, with higher BLH over the dry northern regions (Figure 9b), implying that the hydrologic factors, in addition to solar radiation, may play a role in modulating the spatial distribution of daytime BLH (Seidel et al., 2012).

After sunset, the convective boundary layer undergoes a transition to the nocturnal stable boundary layer. Since the LST is earlier in the west of China than the east (15° longitude is equivalent to a one-hour change in time), the evening transition of boundary layer structure is well manifested by the east-to-west gradient of BLH at 2000 BJT (Figure 8b), although the effects of elevation on the spatial distribution of BLH cannot be ignored. During the nighttime, the BLH is less than 0.3 km over most

China (Figures 10b and 11a). Comparing the BLH at 0200 BJT with that of 1400 BJT, it is found that diurnal amplitude of BLH is stronger in the north and west of China.

   Similar to the spatial distribution of BLH at 2000 BJT, the BLH at 0800 BJT demonstrates the transition from nocturnal boundary layer to convective boundary layer (Figure 7b), characterizing by a west-to-east BLH gradient.

With the ground-based cloud cover observations obtained simultaneously at the same sounding sites, the effects of cloud cover on the diurnal evolution of BLH in summer are investigated as well. As illustrated in Figure 11b, during the daytime the development of PBL is typically suppressed due to the less solar radiation received at the surface under cloudy condition. As a result, at 1400 BJT, the 25th, 50th and 75th percentile values of BLHs decrease by around 0.3 km. And the higher mean and 75th

percentile values of BLHs under clear condition also could be observed at 2000 BJT and 0200 BJT, which may be relevant to the larger heat storage of land surface. In contrast, a slightly thicker boundary





layer could be formed in the early morning under cloudy condition, which may be caused by the stronger atmospheric counter radiation processes.

Using the ERA-BLH as an alternative, the diurnal variations of BLH in the other three seasons are presented in the supplementary materials (Figures S1-3). It is found that the diurnal variation of BLH in spring (Figure S1) is comparable to that of summer, while the diurnal variation of BLH in fall (Figure S2) is less prominent. In winter (Figure S3), due to the weak solar radiation, the development of BLH over the plains of eastern China is severely suppressed, leading to the weakest diurnal variation of BLH during the four seasons over the plain regions.

*4.4 Association of BLHs with related meteorological variables*

As a preliminary effort to explore the potential influence of meteorology on the observed annual cycle of BLH, we turned to ground-based weather observations and soundings that are simultaneously obtained from sites shown in Figure 1. In particular, the association of CMA-BLH with four other atmospheric variables was evaluated using correlation analyses, including surface pressure, 10-m wind speed, near-surface temperature, and lower tropospheric stability (LTS). The LTS is defined as the difference in potential temperature between 700 hPa and the surface pressure (Slingo, 1987), which can be used to describe the thermodynamical state of the lower troposphere (Guo et al., 2016).

For instance, Figure 12 compares the annual cycles of CMA-BLH with these four meteorological variables in Beijing. The annual cycles of CMA-BLHs at 0800 and 2000 BJT are quite similar. In other words, both CMA-BLHs peak in May, and reach the minimum in October. The annual cycle of CMA-





BLH is anti-correlated with the annual cycle of surface pressure (Figure 12a), implying that the seasonal shift of large-scale high/low pressure systems can suppress/facilitate the development of boundary layer through the associated large-scale ascending/descending motion (Liu et al., 2013; Medeiros et al., 2005). The annual cycle of BLH is positively correlated with the annual cycle of near-surface wind speed and temperature (Figures 12b-c). In addition, the annual cycle of CMA-BLH is anti-correlated with the variation of LTS (Figure 12d). Comparing with the near-surface temperature, the variation of LTS is in closer correspondence to the annual cycle of BLH.

Extending this analysis to the full set of radiosonde stations analyzed, Figure 13 presents the correlations among these variables at 0800 and 2000 BJT. The most sites in China are consistent with the results from Beijing. Specially, the annual cycle of CMA-BLHs is anti-correlated with the variation of surface pressure and LTS (Figures 13a, 13b, 13g, and 13h), and positively correlated with near-surface wind speed and temperature (Figures 13c-f). Generally, the correlations are closer at 2000 BJT. And the correlations are weaker in some sites close to the southern coasts and the east of Tibet Plateau, probably due to the modulation of local geographical effects (Stull, 1988).

## 5. Concluding remarks

As the first attempt to obtain climatology of BLHs in China, we have made an extensive use of sounding data from the L-band radiosonde network operated and maintained by CMA from January 2011 to July 2015. The CMA-BLHs were also compared with coincident ERA-BLHs using the same bulk Richardson number method.





The mapping of BLH for the first 41 months of new-generation radiosonde network provides considerable insight into the large-scale spatio-temporal variations of boundary layer characteristics across China with continuous radiosonde data studied to date. Previously unknown facts about boundary layer across whole China were first discovered. A complex pattern of boundary layer has been

revealed, exhibiting strong geographical, diurnal, and seasonal variations, corroborated by BLHs derived from ERA-Interim. There are substantial influences by elevated terrain features and major land–water boundaries. The association of BLH with solar radiation is a compelling relationship when examining the results for the first 4 years of operation of radiosonde network.

The sounding observations at 0800 and 2000 BJT show that the BLH of spring and summer is

generally higher than that of fall and winter. This annual cycle of BLH at most sites is found to be anti-correlated with the annual cycle of surface pressure and LTS, and positively correlated with the near-surface wind speed and temperature. These correlations are stronger at 2000 BJT.

At 2000 BJT, the spatial distribution of CMA-BLH exhibits a pronounced east-west gradient of BLH, and a reversed BLH gradient is presented at 0800 BJT. These spatial patterns of BLH in the early

evening and early morning are likely caused by the different solar radiation in the west (at an earlier Local Sidereal Time) and east of China. The comparison of seasonally averaged CMA-BLH and ERA-BLH at 0800 and 2000 BJT shows good agreement. The seasonal, and spatial variation of CMA-BLH at these two observation times are generally well reproduced by ERA-Interim reanalysis, although the ERA-Interim reanalysis tends to overestimate the BLH in spring and underestimate the BLH in summer.

The climatological diurnal cycle of BLH, revealed by the four-times-daily sounding in summer, shows that BLH peaks in the afternoon and diurnal amplitude of BLH is higher in the north and west of China.





Comparing with other seasons, it is found that the diurnal variation of BLH is typically stronger in spring and summer than in fall and winter.

The climatological features of BLH in China revealed here have important implications for better understanding the formation mechanism of aerosol pollution, cloud and even extreme precipitation in China. In the near future, other algorithms for deriving BLH merit a more comprehensive test based on the radiosonde network of China, apart from further scrutiny for the sources of uncertainty with regard to the data and methods applied in this study.

## Acknowledgements

This study is supported by the National Natural Science Foundation of China under grants 91544217 and 41471301, Ministry of Science and Technology of China under grant 2014BAC16B01, and Chinese Academy of Meteorological Sciences under grant 2014R18. The authors would like to acknowledge CMA for providing the long-term sounding data. Last but not least, we appreciate tremendously the constructive comments offered by the two anonymous reviewers, which greatly improve the quality of this manuscript.

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





# Figures

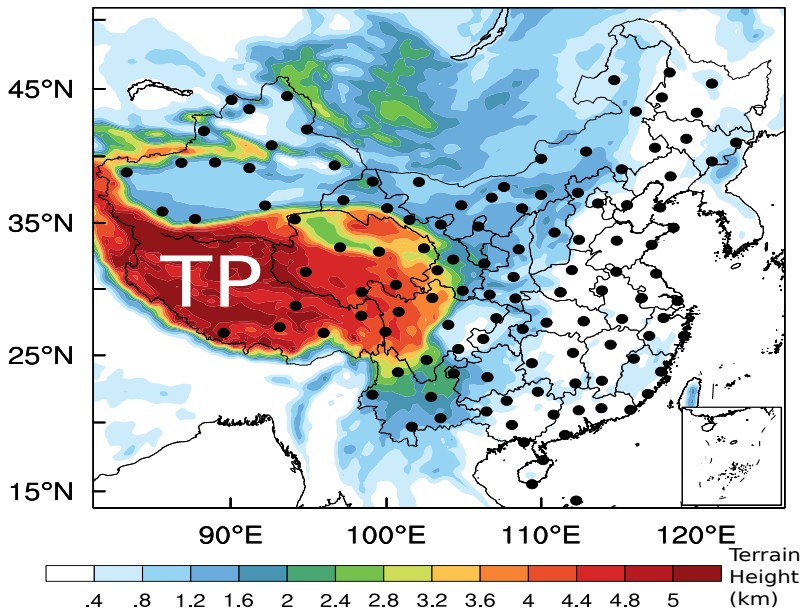

**Figure 1.** Spatial distribution of CMA sounding sites (black dots), overlaid over the

terrain height (color shaded) of China. The white "TP" indicates the location of Tibet

Plateau.





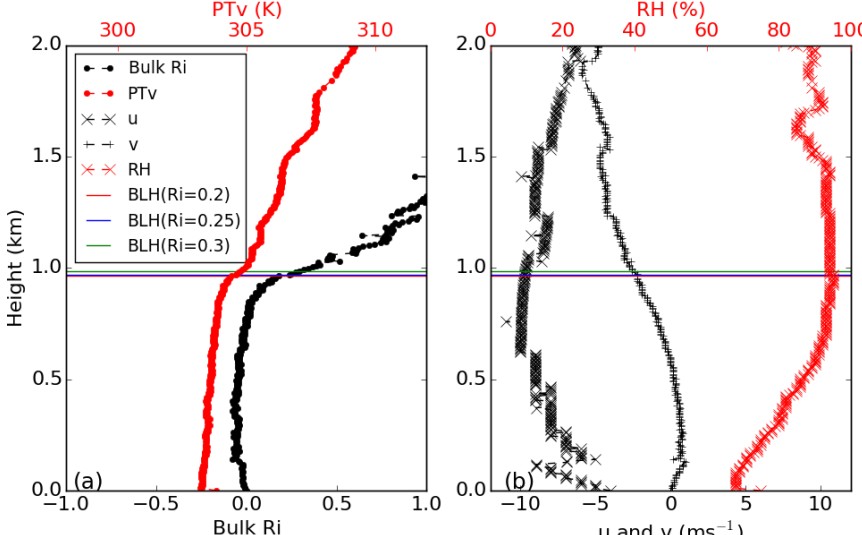

**Figure 2.** Vertical profile of (a) virtual potential temperature (PTv), bulk Richardson

number (Ri), (b) wind speed (U and V), and relative humidity (RH) based on the 2000

BJT 03 August 2012 radiosonde observation at Beijing. The horizontal lines indicate

boundary layer heights (BLH) computed using 0.2 (red), 0.25 (blue), and 0.3 (green)

as critical bulk Richardson number.





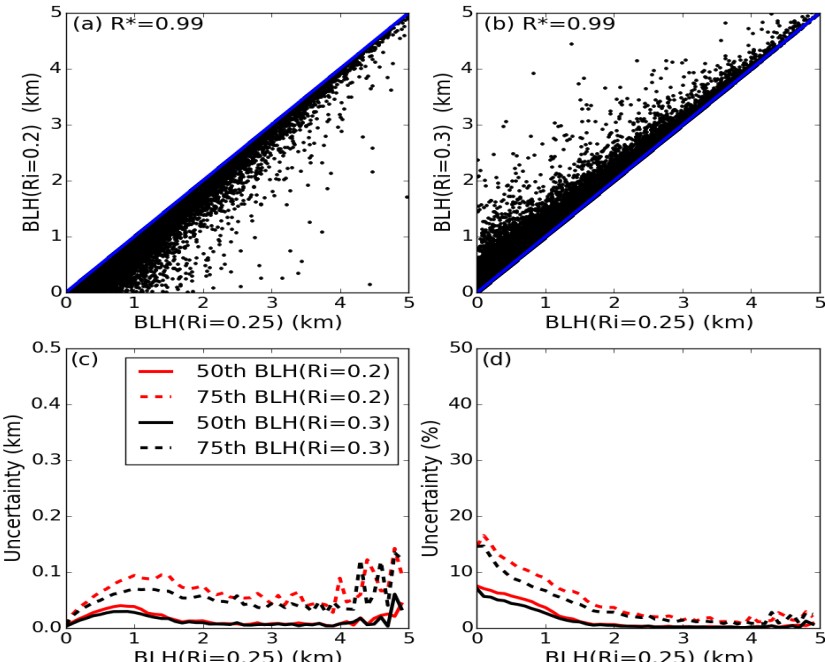

**Figure 3.** Scatter plots showing (a) CMA-BLHs computed using 0.2 and 0.25 as

critical Ri, and (b) CMA-BLHs computed using 0.25 and 0.3 as critical Ri, both of

which are based on 391552 soundings across China from January 2011 to July 2015.

The 50th and 75th percentile values of (c) absolute uncertainty and (d) relative

uncertainty using 0.2 and 0.3 as critical Ri are shown as well. The correlation

coefficients (R) are given in the top panels, where the star superscripts indicate that

values are statically significantly ($p < 0.05$).





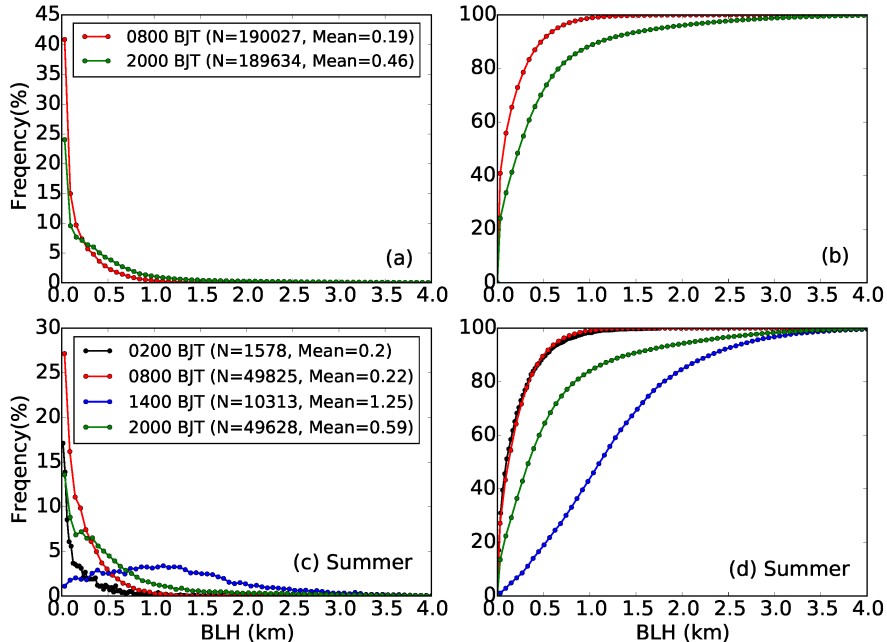

**Figure 4.** Frequency distribution (left) and cumulative frequency distribution (right) of CMA-BLH at 0200 (in black), 0800 (in red), 1400 (in blue) and 2000 BJT (in green) during (a, b) the whole period and (c, d) the summertime from January 2011 to July 2015. The number of soundings (N) and mean value at each observed time are also given. Noted that the soundings at 0200 and 1400 BJT are occasionally launched during certain intensive observation periods in summer.





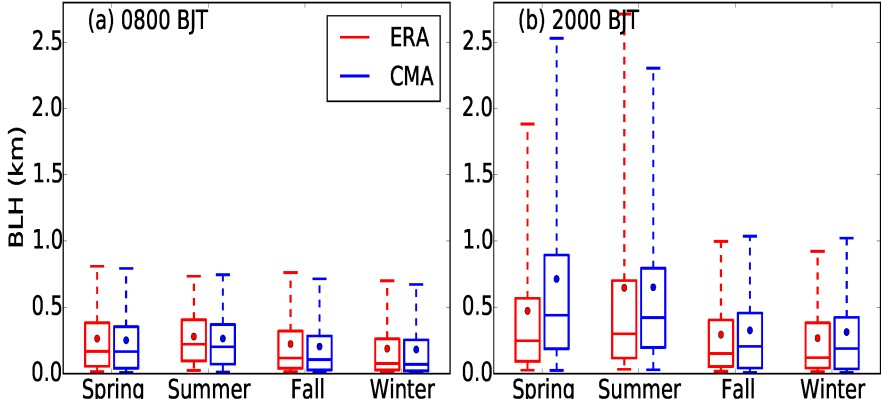

**Figure 5.** Box-and-whiskers plots showing the 5th, 25th, 50th, 75th, and 95th percentile values of ERA-BLH (in red) and CMA-BLH (in blue) during each season at (a) 0800 BJT and (b) 2000 BJT during the period from January 2011 to July 2015. The dot in the each box indicates the mean value of BLHs. Note that there exists larger uncertainty for the evening BLH climatologies in (b).





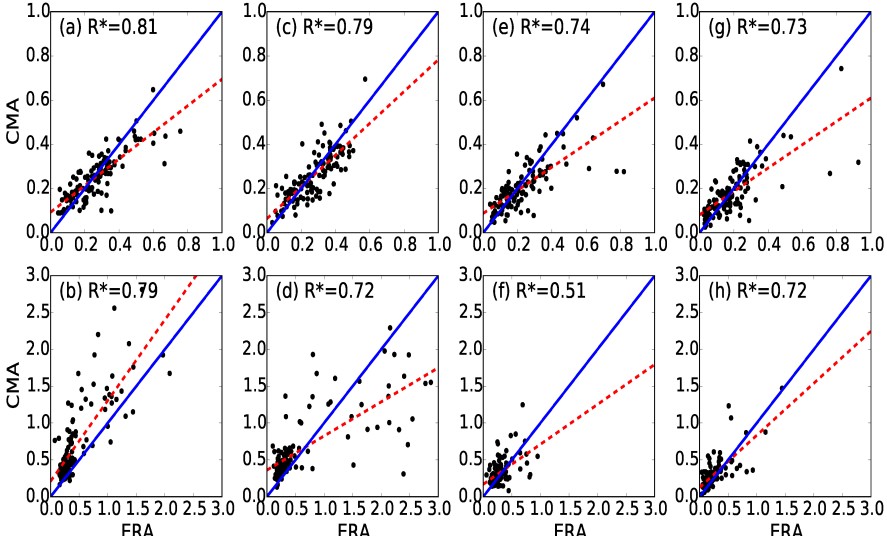

**Figure 6.** Scatter plots of the seasonally averaged CMA-BLH and ERA-BLH at (top) 0800 and (bottom) 2000 BJT in (a, b) spring, (c, d) summer, (e, f) fall, and (g, h) winter. The correlation coefficients (R) are given in each panel, where the star superscripts indicate that values are statically significantly ($p < 0.05$).





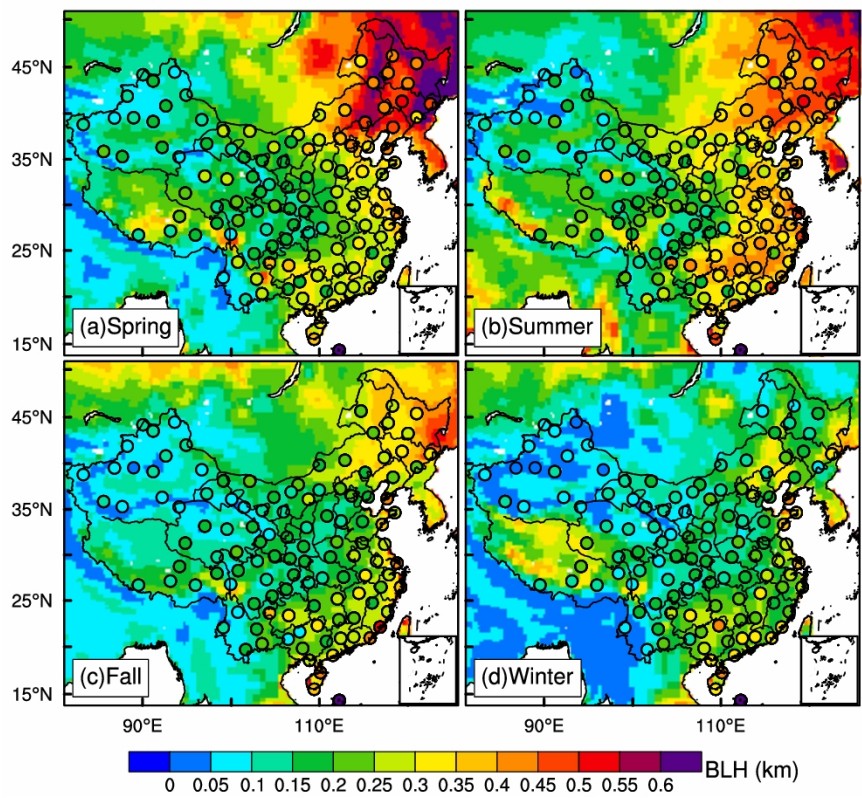

**Figure 7.** Spatial distributions of the seasonal mean of ERA-BLH (color shaded) and

CMA-BLH (color dots) at 0800 BJT in (a) spring, (b) summer, (c) fall, and (d) winter.





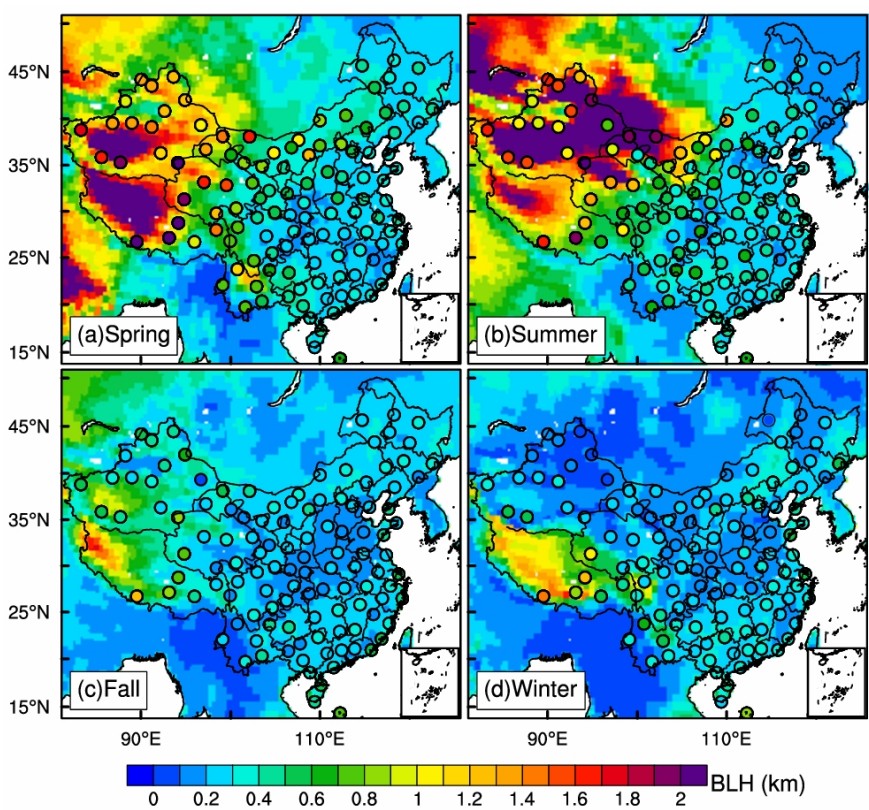

**Figure 8.** Same as in Figure 7, but for the spatial distribution of BLHs at 2000 BJT.





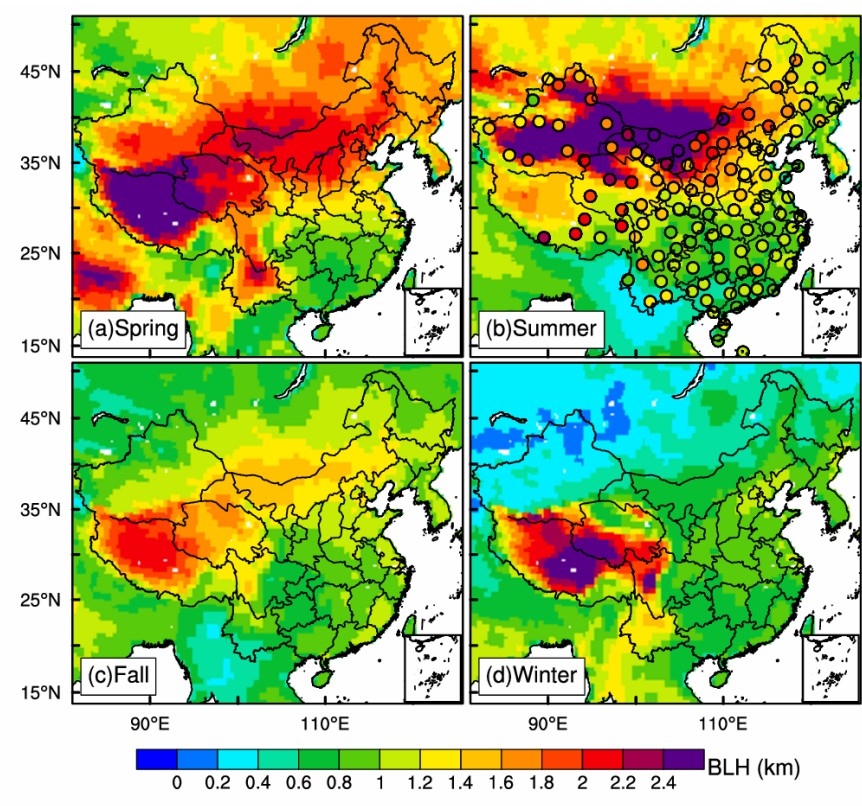

**Figure 9.** Same as Figure 7, but for the spatial distribution of BLHs at 1400 BJT.




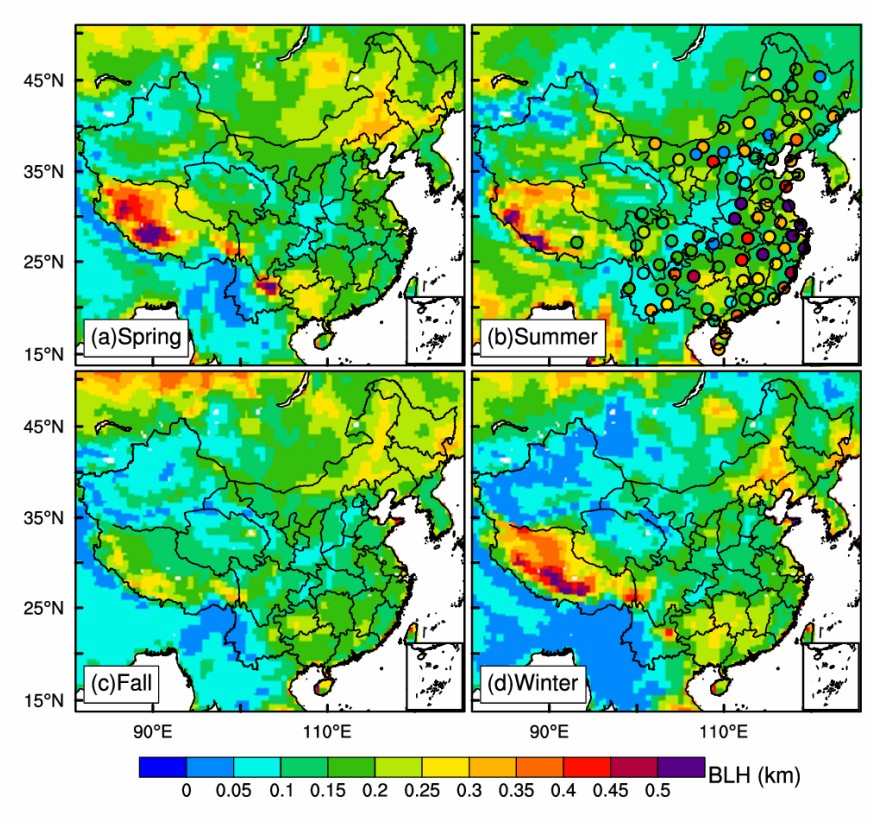

**Figure 10.** Same as in Figure 7, but for the spatial distribution of BLHs at 0200 BJT.





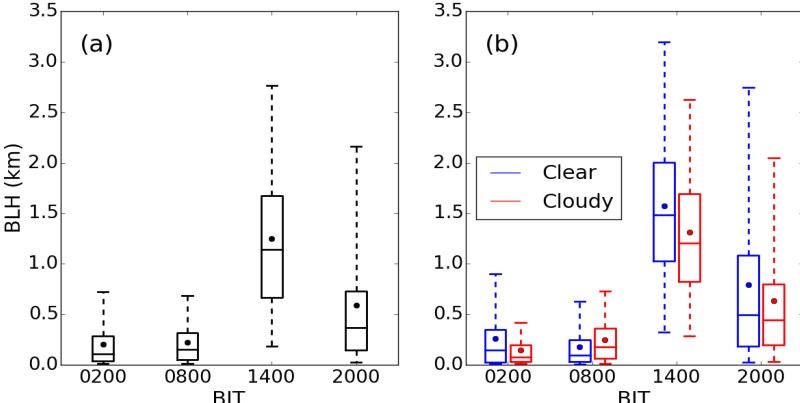

**Figure 11.** (a) Box-and-whiskers plots showing the 5th, 25th, 50th, 75th, and 95th percentile values of CMA-BLH at 0200, 0800, 1400, and 2000 BJT in summer based on all the summertime soundings in China. (b) Box-and-whiskers plots showing CMA-BLH in summer under clear (total cloud cover $\leq$ 20%, in blue) and cloudy (total cloud cover $\geq$ 80%, in red) conditions. Note that the rainy days are not considered in (b).





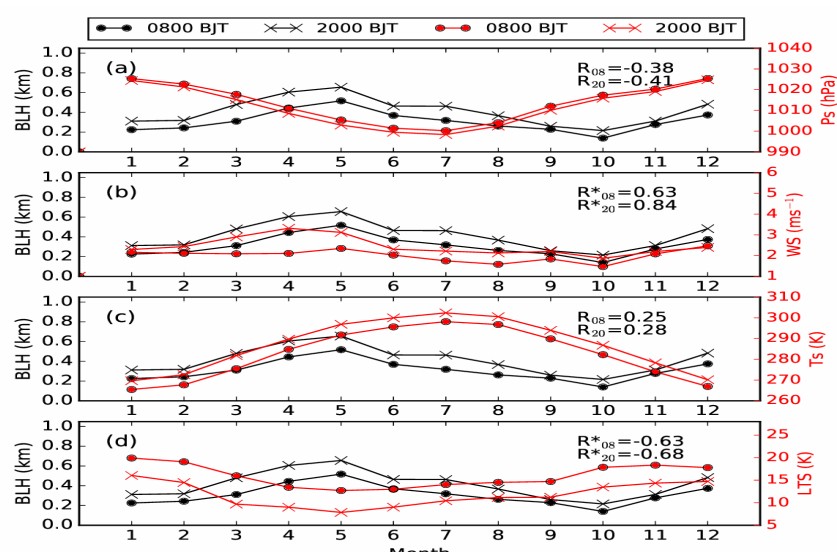

**Figure 12.** Climatological mean annual cycle of CMA-BLH at Beijing (39.8°N, 116.47°E) at 0800 BJT (dot symbols) and 2000 BJT (x symbols), respectively, as superimposed by the counterparts of meteorological parameters, including (a) surface pressure (Ps), (b) 10-m wind speed (WS), (c) near-surface temperature (Ts), and (d) Lower Tropospheric Stability (LTS). The correlation coefficients at 0800 ($R_{08}$) and 2000 BJT ($R_{20}$) between of CMA-BLH and meteorological parameters are given in each panel as well, and the star superscripts indicate the values that are statistically significant ($p < 0.05$).





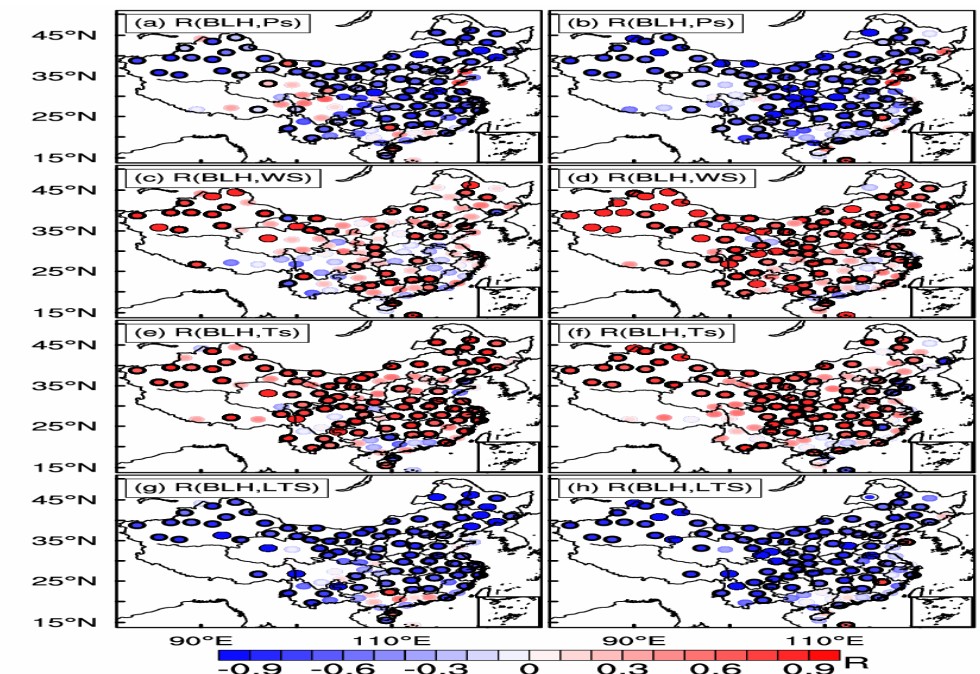

**Figure 13.** Correlations (R) between mean annual cycles (12 monthly values) of CMA-BLH and (a, b) surface pressure (Ps), (c, d) 10-m wind speed, (e, f) near-surface temperature (Ts), (g, h) Lower Tropospheric Stability (LTS), at (left) 0800 and (right) 2000 BJT. Symbols outlined in black indicate values that are statistically significant ($p < 0.05$)