# Peer review of "The climatology of planetary boundary layer height in China derived from radiosonde and reanalysis data"

_Atmospheric Chemistry and Physics, 2016_

## Referee Comment (RC1) · Anonymous Referee #1 · 29 Jul 2016

The manuscript reports climatology of planetary boundary layer (PBL) height in China from sounding observations and a reanalysis data, and explores the relationship between PBL height and meteorological factors such as surface pressure, wind speed, atmospheric instability, etc. As this study represents the first effort to compile the historical records of sounding-based PBL climatology in China, the results in this paper and future exploration of the data are crucial for understanding the regional climate changes in East Asia. I recommend accepting the manuscript by ACP after the authors address some minor comments below.

1) Can the authors provide the related equations that estimate BLH using the bulk Ri method?

2) Page 9, line 15-20. The authors stated that PBL height is higher in spring than that in summer and they attributed it to the stronger near-surface wind spring. To me, such a seasonal difference of PBL is only obvious at 2000 BJT, not for 0800 BJT, according to Fig. 5. So my question is if the wind only differs at 2000 BJT between spring and summer? An additional plot may help to elucidate it.

3) Fig. 7. Tibet Plateau shows the highest PBL height in winter. Why?

4) Fig. 13. How about the correlations based on monthly mean data for each month other than the annual cycles? Meanwhile, the plot needs to be stretched vertically.

5) The interference between PBL height and atmospheric aerosol concentrations were widely discusses by previous studies [Y. Wang et al., Atmos. Env., 2013; R. Zhang et al., Chem. Rev., 2015], especially over China [A. Ding et al., GRL, 2016]. Those discussions are worthy in the introduction part to emphasize the importance of the PBL record in the air pollution research.

———————————————————

---

## Referee Comment (RC2) · Anonymous Referee #2 · 1 Aug 2016

[referee-annotated manuscript omitted]

---

## Referee Comment (RC3) · Anonymous Referee #3 · 1 Aug 2016

Guo et al. present a well written manuscript that reports on a comprehensive investigation of the planetary boundary layer height (BLH) in China using fine-resolution sounding observations and reanalysis data from January 2011 to July 2015. The bulk Richardson number based BLH estimation method is a popular approach and the results are finely analyzed and discussed. The findings are important in understanding air pollution in China. This manuscript should be published after my suggestions below have been satisfactorily addressed.

My full comments are included as a supplement pdf.

Please also note the supplement to this comment:

[Figure]

http://www.atmos-chem-phys-discuss.net/acp-2016-564/acp-2016-564-RC3-supplement.pdf

[Figure]

**Supplement:**

Guo et al. present a well written manuscript that reports on a comprehensive investigation of the planetary boundary layer height (BLH) in China using fine-resolution sounding observations and reanalysis data from January 2011 to July 2015. The bulk Richardson number based BLH estimation method is a popular approach and the results are finely analyzed and discussed. The findings are important in understanding air pollution in China. This manuscript should be published after my suggestions below have been satisfactorily addressed.

**Specific Comments:**

1. The physical meaning of the Richardson number method, turbulent flow becomes laminar flow when $Ri > R_C$ (Stull, 1988), should be included in the manuscript in a proper way. $R_C = 0.25$ is the critical value of the bulk Richardson number.

2. Which map projection is used in the manuscript (Figs. 1, 7-10, 13)?

3. Page 5, Line 10: "121operational" $\rightarrow$ "121 operational".

4. Page 6, Line 10: "0.7 degree" $\rightarrow$ "0.75 degree"?

5. Page 7, Line 7: Add latitude, longitude and altitude for Beijing site.

6. Page 8, Line 4: "Figures 2" $\rightarrow$ "Figure 2".

7. Page 10, Lines 16-17: "due to the solar radiation in eastern China is stronger than that of western regions" $\rightarrow$ "due to stronger solar radiation in eastern China than that of western regions".

8. Page 14, Line 17: It is not appropriate to write "As the first attempt to obtain climatology of BLHs in China" because Liu et al. (2015) have already done research on BLH over China using CALIPSO and ERA-Interim reanalysis data.

9. Page 16, Line 17: "Reference" $\rightarrow$ "References".

10. Page 31, Fig. 10: ERA-BLH and CMA-BLH don't agree well at 0200 BJT, why?

**References**

Liu, J., Huang, J., Chen, B., Zhou, T., Yan, H., Jin, H., Huang, Z., and Zhang, B.: Comparisons of PBL heights derived from CALIPSO and ECMWF reanalysis data over China, Journal of Quantitative Spectroscopy and Radiative Transfer, 153, 102-112, doi:http://dx.doi.org/10.1016/j.jqsrt.2014.10.011, 2015.

Stull, R. B.: An Introduction to Boundary Layer Meteorology, edited by R. B. Stull, Springer 5

Netherlands, Dordrecht., 1988.

---

## Author Comment (AC1) · 5 Sep 2016

The manuscript reports climatology of planetary boundary layer (PBL) height in China from sounding observations and a reanalysis data, and explores the relationship be- tween PBL height and meteorological factors such as surface pressure, wind speed, atmospheric instability, etc. As this study represents the first effort to compile the historical records of sounding-based PBL climatology in China, the results in this paper and future exploration of the data are crucial for understanding the regional climate changes in East Asia. I recommend accepting the manuscript by ACP after the authors address some minor comments below.

*Response:We are quite grateful to referee #1 for his/her positive comments on our work, which are quite constructive and helpful. All these comments and concerns raised by the referee have been explicitly considered and incorporated into this revision. For clarity purpose, here we have listed the reviewers' comments in plain font, followed by our responses in italics.*

1) Can the authors provide the related equations that estimate BLH using the bulk Ri method?

*Response: Per your suggestion, the following descriptions with regard to the physical meaning in Equation (1) were added in the revised manuscript:*

*"Note that the Ri is dimensionless, and has nothing to do with the intensity of turbulence, but it can tell whether the turbulence exists or not. Previous theoretical and laboratory studies (e.g., Stull, 1988) suggested that when Ri is smaller than the critical value (~0.25), the laminar flow becomes unstable. Thus, the lowest level z at which interpolated Ri crosses the critical value of 0.25 is referred to as BLH in this study, similar to the criteria used by Seidel et al. (2012)."*

2) Page 9, line 15-20. The authors stated that PBL height is higher in spring than that in summer and they attributed it to the stronger near-surface wind spring. To me, such a seasonal difference of PBL is only obvious at 2000 BJT, not for 0800 BJT, according to Fig. 5. So my question is if the wind only differs at 2000 BJT between spring and summer? An additional plot may help to elucidate it.

*Response: The development of PBL is primarily determined by the buoyancy and mechanical forcing (Stull 1988). The strengths of buoyancy and mechanical forcing are relevant to the near-surface thermal stability and wind speed, respectively. Either strong near-surface wind speed or weak stability favors the development of PBL. To*

*better address your concerns, we made the following plots (Figs. R1-R4). The near-surface wind speed in spring is stronger than that of summer at both 0800 and 2000 BJT at most sites (Figs. R1-R2).*

*At 0800 BJT, although the near-surface wind speed in spring is stronger than that in summer (Fig.R1), the near-surface stability is also stronger than that of summer, especially for the eastern China (Fig. R3). Due to these contrary influences of mechanical forcing and stability, at 0800 BJT the BLH of spring is not generally higher than that of summer.*

*At 2000 BJT, the near-surface wind speed of spring is stronger than that of summer (Fig. R2), and the near-surface thermal stabilities of these two seasons are comparable (Fig. R4). Thus, at 2000 BJT the BLH of spring is generally higher than that of summer. As a result, further discussions have been added in revised Section 4.2.*

[Figure]

*Figure R1. Spatial distribution of near-surface wind speed (a) in spring at 0800 BJT, and the wind speed difference (subtracted from that of spring) in (b) summer, (c) fall, and (d) winter. The seasonally averaged near-surface wind speed was obtained from the CMA soundings by averaging the observations below 100 m AGL.*

[Figure]

*Figure R2. Similar as Fig.R1, but for the spatial distribution of near-surface wind speed at 2000 BJT.*

[Figure]

*Figure R3. Spatial distribution of near-surface inversion strength (vertical potential temperature gradient between 10 m and 100 m AGL) at 0800 BJT in (a) spring, (b) summer, (c) Fall, (d) Winter.*

[Figure]

*Fig.R4. Similar as Fig.R3, but for the spatial distribution of inversion strength at 2000 BJT.*

3) Fig. 7. Tibet Plateau shows the highest PBL height in winter. Why?

*Response: As the reviewer pointed out, at 0800 BJT (~0600 LST) the highest ERA-BLH over the northern Tibet Plateau (TP) region occurs in winter, which may be caused by the strongest near-surface wind in winter over the region (Fig. R5), which was based on reanalysis data.*

[Figure]

*Figure R5. Spatial distribution of seasonally averaged 10-m wind speed in (a) spring, (b) summer, (c) fall, (d) winter derived from ERA-Interim reanalysis from January 2011 to July 2015.*

4) Fig. 13. How about the correlations based on monthly mean data for each month other than the annual cycles? Meanwhile, the plot needs to be stretched vertically.

*Response: Per your suggestion, we compared the monthly mean values of CMA-BLH with the four meteorological parameters, including surface pressure, 10-m wind speed, near-surface temperature, and lower tropospheric stability (Figs. R6 and R7). Although the correlation coefficients for a specific site are different (Fig.R6), the correlations (Fig. R7) on the whole are generally similar to those of annual cycles (Fig. 13 in the revised manuscript). Namely, the BLH is positively correlated with the near-surface wind speed and temperature, and anti-correlated with the lower tropospheric stability and surface pressure.*

*By the way, Fig. 13 you pointed out has been stretched vertically, and relevant modifications have been added in the revised Section 4.4.*

[Figure]

*Figure R6. Monthly mean of CMA-BLH at Beijing (39.8 $^o$N, 116.47 $^o$E) from January 2011 to July 2015 at 0800 BJT (dot symbols) and 2000 BJT (x symbols), respectively, as superimposed by the counterparts of meteorological parameters, including (a) surface pressure (Ps), (b) 10-m wind speed (WS), (c) near-surface temperature (Ts), and (d) Lower Tropospheric Stability (LTS). The correlation coefficients at 0800 (R08) and 2000 BJT (R20) between of CMA-BLH and meteorological parameters are given in each panel as well, and the star superscripts indicate the values that are statistically significant (p < 0.05).*

[Figure]

*Figure R7. Correlations (R) between monthly mean values (from January 2011 to July 2015) of CMA-BLH and (a, b) surface pressure (Ps), (c, d) 10-m wind speed, (e, f) near-surface temperature (Ts), (g, h) Lower Tropospheric Stability (LTS), at (left) 0800 and (right) 2000 BJT. Symbols outlined in black indicate values that are statistically significant (p < 0.05)*

5) The interference between PBL height and atmospheric aerosol concentrations were widely discusses by previous studies [Y. Wang et al., Atmos. Env., 2013; R. Zhang et al., Chem. Rev., 2015], especially over China [A. Ding et al., GRL, 2016]. Those discussions are worthy in the introduction part to emphasize the importance of the PBL record in the air pollution research

*Response: Per your suggestion, the references have been added in the introduction section to highlight the importance of these studies.*

.

---

## Author Comment (AC2) · 5 Sep 2016

**Anonymous Reviewer #2:**

This paper investigates the planetary boundary layer height over many regions in China on a seasonal and diurnal basis. It shows the non-uniformity of the BLH over the country and uses high-resolution radiosonde observations and reanalysis data. Overall this is a good study and fits the scope of ACP and suggestions for revisions have been provided in a supplemental PDF.

*Response*:*We are quite grateful to referee #2 for his/her positive comments on our manuscript, which are pretty constructive and valuable. Because all these suggestions for revisions raised by the referee are almost all grammar errors or typos, which have been explicitly considered and incorporated into this revision, we will not list here the response one-by-one to each comment or suggestion.*

---

## Author Comment (AC3) · 5 Sep 2016

**Anonymous Reviewer #3:**

Guo et al. present a well written manuscript that reports on a comprehensive investigation of the planetary boundary layer height (BLH) in China using fine-resolution sounding observations and reanalysis data from January 2011 to July 2015. The bulk Richardson number based BLH estimation method is a popular approach and the results are finely analyzed and discussed. The findings are important in understanding air pollution in China. This manuscript should be published after my suggestions below have been satisfactorily addressed.

*Response:We are quite grateful to referee #3 for his/her positive comments on our work, which are quite constructive and helpful. All these comments and concerns raised by the referee have been explicitly considered and incorporated into this revision. For clarity purpose, here we have listed the reviewers' comments in plain font, followed by our response in italics.*

**Specific Comments:**

1. The physical meaning of the Richardson number method, turbulent flow becomes laminar flow when $Ri > Rc$ (Stull, 1988), should be included in the manuscript in a proper way. $Rc = 0.25$ is the critical value of the bulk Richardson number.

*Response: The following paragraph was added in the method section of this revised manuscript to better describe the physical meaning of the bulk Ri method:*

*"Note that the Ri is dimensionless, and itself says nothing about the intensity of turbulence, but it can tell whether the turbulence exists or not; theoretical and laboratory research suggest that when Ri is smaller than the critical value (~0.25), the laminar flow becomes unstable (Stull, 1988). Thus, the lowest level z at which interpolated Ri crosses the critical value of 0.25 is referred to as BLH in this study, similar to the study of Seidel et al. (2012)."*

2. Which map projection is used in the manuscript (Figs. 1, 7-10, 13)?

*Response: The map project we used is the Lambert-conformal conic projection.*

3. Page 5, Line 10: "121operational" → "121 operational".

*Response: Amended as suggested.*

4. Page 6, Line 10: "0.7 degree" → "0.75 degree"?

*Response: Amended as suggested.*

5. Page 7, Line 7: Add latitude, longitude and altitude for Beijing site.

*Response: "(116.47$^o$E, 39.80$^o$N, 32m a.s.l.)" has been added for Beijing site.*

6. Page 8, Line 4: "Figures 2" → "Figure 2".

*Response: Amended as suggested.*

7. Page 10, Lines 16-17: "due to the solar radiation in eastern China is stronger than that of western regions" → "due to stronger solar radiation in eastern China than that of western regions".

*Response: Corrected.*

8. Page 14, Line 17: It is not appropriate to write "As the first attempt to obtain climatology of BLHs in China" because Liu et al. (2015) have already done research on BLH over China using CALIPSO and ERA-Interim reanalysis data.

*Response: Per your suggestion, it has been revised to "As the first effort to compile the climatological sounding-based BLHs in China...". Meanwhile, Liu et al., 2015 has been cited in the section of Introduction.*

9. Page 16, Line 17: "Reference"→ "References".

*Response: Amended as suggested.*

10. Page 31, Fig. 10: ERA-BLH and CMA-BLH don't agree well at 0200 BJT, why?

*Response: To our knowledge, the soundings at 0200 BJT were launched occasionally in summer in attempt to capture the major severe storms taking place in China, and*

*consequently a total of 1,578 profiles were collected and used for this study. As shown in Fig. R8, most stations have less than 30 of sounding profiles. Due to this sampling limitation, the seasonally averaged CMA-BLH at 0200 BJT don't agree well with ERA-BLH. Therefore, to reflect this issue, the original Fig. 10 was replaced by Fig. R8 in the revised manuscript, in which only the stations have more than 30 soundings are plotted.*

[Figure]

*Figure R8. Spatial distributions of the seasonal mean of ERA-BLH (color shaded) and CMA-BLH (color dots) at 0200 BJT in (a) spring, (b) summer, (c) fall, and (d) winter. Note that only the stations have more than 30 soundings are plotted in (b).*

**References**

Liu, J., Huang, J., Chen, B., Zhou, T., Yan, H., Jin, H., Huang, Z., and Zhang, B.: Comparisons of PBL heights derived from CALIPSO and ECMWF reanalysis data over China, Journal of Quantitative Spectroscopy and Radiative Transfer, 153, 102-112, doi:http://dx.doi.org/10.1016/j.jqsrt.2014.10.011, 2015.

Stull, R. B.: An Introduction to Boundary Layer Meteorology, edited by R. B. Stull, Springer 5 Netherlands, Dordrecht, 1988.

---

## Author Response (AR2)

**Anonymous Reviewer #2:**

There are a few more minor suggestions for this manuscript. On p. 5 line 5-10, the authors discussed the interaction between PBL and pollution development. It would be desirable to add some observation
5 evidences for such an interaction. For example, Guo et al. 2014 [Elucidating severe urban haze formation in China, Proc. Natl. Acad. Sci. USA 111, 17373] found a lack of a diurnal variation, but a cycle of 4-7 days in the aerosol properties, indicating a reduced PBL diurnal trend during polluted periods. Furthermore, another recent measurement [Peng et al., 2016, Markedly enhanced absorption and direct radiative forcing of black carbon under polluted urban environments, Proc. Natl. Acad. Sci.
10 USA 113, 4266] has demonstrated rapid aging of black carbon particles in China and considerably enhanced light absorption by aged black carbon particles, suggesting a large impact of atmospheric stabilization by black carbon particles. The effects of black carbon particles on PBL have also been adequately discussed in an early important publication [Zhang et al., 2008, Variability in morphology, hygroscopic and optical properties of soot aerosols during internal mixing in the atmosphere, Proc. Natl.
15 Acad. Sci. USA 105, 10291].

*Response:We are quite grateful to referee #2 for his/her constructive comments on our work. Per the suggestion, we added the following sentences in the revised manuscript.*

[revised manuscript text omitted]